RESOURCE REPORT

# A Leak-Free Inducible CRISPRi/a System for Gene Functional Studies in *Plasmodium falciparum*

Xiaoying Liang,[a] Rachasak Boonhok,[a] Faiza Amber Siddiqui,[a] Bo Xiao,[b] Xiaolian Li,[a] Junling Qin,[a] Hui Min,[a] Lubin Jiang,[b] Liwang Cui,[a,c] Jun Miao[a,c]

[a]Department of Internal Medicine, Morsani College of Medicine, University of South Florida, Tampa, Florida, USA

[b]Unit of Human Parasite Molecular and Cell Biology, Key Laboratory of Molecular Virology and Immunology, Pasteur Institute of Shanghai, Chinese Academy of Sciences, Shanghai, People's Republic of China

[c]Center for Global Health and Infectious Diseases Research, College of Public Health, University of South Florida, Tampa, Florida, USA

**ABSTRACT** By fusing catalytically dead Cas9 (dCas9) to active domains of histone deacetylase (Sir2a) or acetyltransferase (GCN5), this CRISPR interference/activation (CRISPRi/a) system allows gene regulation at the transcriptional level without causing permanent changes in the parasite genome. However, the constitutive expression of dCas9 poses a challenge for studying essential genes, which may lead to adaptive changes in the parasite, masking the true phenotypes. Here, we developed a leak-free inducible CRISPRi/a system by integrating the DiCre/*loxP* regulon to allow the expression of dCas9-GCN5/-Sir2a upon transient induction with rapamycin, which allows convenient transcriptional regulation of a gene of interest by introducing a guide RNA targeting its transcription start region. Using eight genes that are either silent or expressed from low to high levels during asexual erythrocytic development, we evaluated the robustness and versatility of this system in the asexual parasites. For most genes analyzed, this inducible CRISPRi/a system led to 1.5- to 3-fold up-or downregulation of the target genes at the mRNA level. Alteration in the expression of *PfK13* and *PfMYST* resulted in altered sensitivities to artemisinin. For autophagy-related protein 18, an essential gene related to artemisinin resistance, a >2-fold up- or downregulation was obtained by inducible CRISPRi/a, leading to growth retardation. For the master regulator of gametocytogenesis, *PfAP2-G*, a >10-fold increase of the *PfAP2-G* transcripts was obtained by CRISPRa, resulting in >4-fold higher gametocytemia in the induced parasites. Additionally, inducible CRISPRi/a could also regulate gene expression in gametocytes. This inducible epigenetic regulation system offers a fast way of studying gene functions in *Plasmodium falciparum*.

**IMPORTANCE** Understanding the fundamental biology of malaria parasites through functional genetic/genomic studies is critical for identifying novel targets for antimalarial development. Conditional knockout/knockdown systems are required to study essential genes in the haploid blood stages of the parasite. In this study, we developed an inducible CRISPRi/a system via the integration of DiCre/*loxP*. We evaluated the robustness and versatility of this system by activating or repressing eight selected genes and achieved up- and downregulation of the targeted genes located in both the euchromatin and heterochromatin regions. This system offers the malaria research community another tool for functional genetic studies.

**KEYWORDS** human malaria, *Plasmodium falciparum*, dCas9, gene regulation, DiCre

Despite tremendous efforts in controlling malaria, malaria still accounted for an estimated 229 million cases and 409,000 deaths globally in 2019, according to the World Malaria Report 2020. The emergence and spread of *Plasmodium falciparum* resistance to artemisinin-combination therapies have created an obstacle to malaria control and

Address correspondence to Jun Miao, jmiao1@usf.edu.

The authors declare no conflict of interest.

elimination. Therefore, a better understanding of the fundamental biology of malaria parasites via functional genomic studies is critical for identifying novel targets for antimalarial development and elucidating the drug resistance mechanism. However, genetic manipulation in *P. falciparum* relies exclusively on homologous recombination because *Plasmodium* lacks RNA interference (RNAi) and the canonical nonhomologous end-joining (NHEJ) repair mechanism (1, 2). Thus, essential genes for blood-stage development cannot be knocked out conventionally since blood-stage parasites are haploid. To overcome this limitation, several conditional genetic modification (knockdown or knockout) systems have been developed to regulate genes of interest at the DNA, transcript, or protein level. For example, the inducible Cre recombinase (DiCre), which recognizes the *loxP* sites, catalyzes the excision or inversion of the flanked DNA fragment upon treatment with rapamycin (Rap) (3–5). The TetR-DOZI and *glmS* ribozyme systems regulate gene expression at the posttranscription level (6–8). The TetR-DOZI system regulates the translation of the target mRNA by binding TetR-DOZI to the TetR aptamers inserted in the 3′ or 5′ untranslated region (UTR) of the target mRNA, blocking its translation and promoting mRNA degradation. The addition of anhydrotetracycline (aTc) removes the TetR-DOZI protein from the aptamers, allowing protein translation (8, 9). The *glmS* ribozyme is inserted in the 3′ UTR of a target gene, and it cleaves the target mRNA upon induction with glucosamine, leading to a reduction in mRNA expression (6). To regulate protein expression at the protein level, an engineered version of the human FKBP12 is fused to the protein of interest, targeting it to degradation, while the fusion protein is stabilized by the cell-permeable synthetic molecule Shield 1 (10–12).

The discovery of the CRISPR/Cas9 system has revolutionized genome editing in a large number of organisms (13, 14). Its applications to malaria parasites have dramatically expanded the toolkits for functional studies, allowing genes to be tagged, disrupted, or deleted and point mutations to be generated (15–17). With this system, the Cas9 endonuclease binds to a single guide RNA (sgRNA), which contains the Cas9 binding domain and 20 customizable nucleotides complementary to a target DNA site. The protospacer adjacent motif (PAM), a sequence immediately downstream from the target region, is required for recruiting Cas9 to the target site to cause a double-strand break (DSB). Depending on the organisms, this break is repaired by homology-directed repair (HDR) using donor DNA or the error-prone NHEJ, causing gene mutation, insertion, or deletion with or without genetic scarring (13–18). In addition to gene editing, this system has been systematically repurposed using the catalytically dead Cas9 (dCas9) for epigenetic editing referred to as CRISPR interference/activation (CRISPRi/a) by tethering an activator, silencer, or modifier of histone/DNA to the dCas9 (19–25). With this modification, dCas9, guided by sgRNA, binds to the promoter/enhancer region with the same efficiency as the active Cas9 but cannot generate a DSB, resulting in the blockade of the transcriptional process or alteration of epigenetic activities of the neighboring target genes. CRISPRi has been successfully adapted in *P. falciparum* and *Plasmodium yoelii* to knock down gene expression by simply guiding dCas9 alone to the region upstream of the gene of interest (26, 27). Xiao et al. designed a CRISPRi/a system by fusing dCas9 with either the histone acetyltransferase (HAT) domain of PfGCN5 or the histone deacetylase (HDAC) domain of PfSir2a, a class III histone deacetylase, to epigenetically modify the promoter chromatin and achieve upregulation or downregulation of the target gene in *P. falciparum* (28). However, with constitutive expression of the dCas9, this system may present a challenge for studying certain essential genes, whose altered expression resulting from the dCas9 system may be too deleterious to allow parasite growth. In addition, constant gene activation or silencing may cause adaptation of the parasites and mask the true phenotype.

To solve the potential problem associated with the constant expression of dCas9, we attempted to incorporate the inducibility of dCas9 expression into this system. We established two inducible dCas9 expression modules using the TetR-DOZI and the DiCre/*loxP* systems and selected the latter for leak-free inducible expression of dCas9-GCN5 and dCas9-Sir2a. We implemented the inducible expression of dCas9-GCN5 and

dCas9-Sir2a to achieve targeted up- and downregulation of genes located in both the euchromatin and heterochromatin regions.

## RESULTS

**Generation of the inducible CRIPRi/a systems.** CRISPR-guided dCas9 fused with the HAT or HDAC creates a valuable tool to regulate target gene expression in *P. falciparum* (28). To introduce inducibility into this system, we attempted the TetR-DOZI-based gene knockdown and the DiCre-based conditional deletion. To conditionally activate the expression of dCas9-GCN5 or -Sir2a using the TetR-DOZI regulon, we insert the 10 aptamer repeats in the 3′ UTR of the dCas9-GCN5/-Sir2a expression cassette (Fig. S1A). This would lead to translation inhibition of the dCas9-fusion protein in the absence of aTc and induced expression of the fusion protein when aTc is added in culture (Fig. S1A). Immunofluorescence assay (IFA) did not detect fluorescent signals when the parasites were cultured without aTc. Thirty hours after adding 0.5 $\mu$M aTc to the synchronized ring-stage parasites, robust fluorescent signals were detected in the nuclei of the parasites, indicating dCas9-fusion protein expression (Fig. S1B). Western blotting further confirmed the induced expression of the fusion proteins after adding aTc for 30 h (Fig. S1C). However, we also detected low-level dCas9-fusion protein expression in the absence of aTc (Fig. S1C), suggesting leaky expression of the fusion protein, which agrees with published studies (8, 29, 30).

To build a DiCre-based induction system, we inserted a green fluorescent protein (GFP) expression cassette between the *hsp86* promoter and the dCas9-GCN5 or dCas9-Sir2a open reading frame. The inclusion of a stop codon at the end of GFP and the *hsp86* 3′ UTR sequence allowed GFP expression but prevented the expression of the downstream dCas9-GCN5 or -Sir2a. Two *loxP* sites were inserted before the GFP start codon and the end of *hsp86* 3′ UTR to allow inducible excision of the *loxP*-flanked fragment via activation of the DiCre recombinase by rapamycin (Rap), resulting in the inducible expression of dCas9-GCN5 and dCas9-Sir2a (Fig. 1A). The transgenic parasite lines are named *loxPed* GFP-dCas9-GCN5 and -Sir2a. These constructs were transfected into the *P. falciparum* clone constitutively expressing DiCre (5) to obtain the transgenic parasite lines. As expected, GFP was ubiquitously expressed in the parasites (Fig. 1B, upper panel), whereas no dCas9-GCN5 or Cas9-Sir2A was detected by Western blotting using the anti-Cas9 antibodies before Rap induction (Fig. 1B, lower panel).

To monitor the DiCre recombinase excision dynamics and efficiency induced by Rap, parasites at the early ring stage were treated with 100 nM Rap for 2 h, and parasite genomic DNA was isolated at 0, 6, 12, 24, and 48 h. Excision of the *gfp-hsp86* 3′ UTR fragment was identified by PCR amplification of the *loxP* locus before excision (1,785 bp) and after excision (122 bp). PCR analysis showed that the *loxP* excision level and excised/unexcised ratio increased over time (Fig. 1C and D). Concomitantly, Western blotting confirmed the declining expression of GFP and gradually increased expression of the dCas9 fusion protein, with the dCas9 fusion protein level reaching the highest at the schizont stage (Fig. 1E). Live-cell fluorescence microscopy and Western blotting also detected the disappearance of the GFP fluorescent signals and the appearance of dCas9 fusion proteins 48 h after Rap treatment (Fig. 1B). Similar DiCre recombinase excision dynamics starting at the trophozoite and schizont stage were detected, indicating that excision is not stage-specific (Fig. S2). Additionally, there was no noticeable change in the parasite growth and developmental progression before and after Rap induction compared to those of the wild-type parasites. These results indicated that the expression of dCas9-GCN5/Sir2a in the DiCre system was leak free and could be robustly induced in the same intraerythrocytic developmental cycle (IDC).

**Regulation of gene expression using the DiCre/*loxP*-inducible dCas9 system.** We first tested the DiCre system as a leak-free inducible dCas9-GCN5/Sir2a system for the proof-of-principle regulation of eight genes that are expressed at high (*PfGCN5*), medium (clathrin heavy chain: *PfCHC, PfK13, PfATG18,* and *PfCRT*), and low (*PfMYST* and *PfDNMT*) or silent (*PfAP2-G*) during the IDC (31) (Fig. 2A, Table S1). Of the eight genes

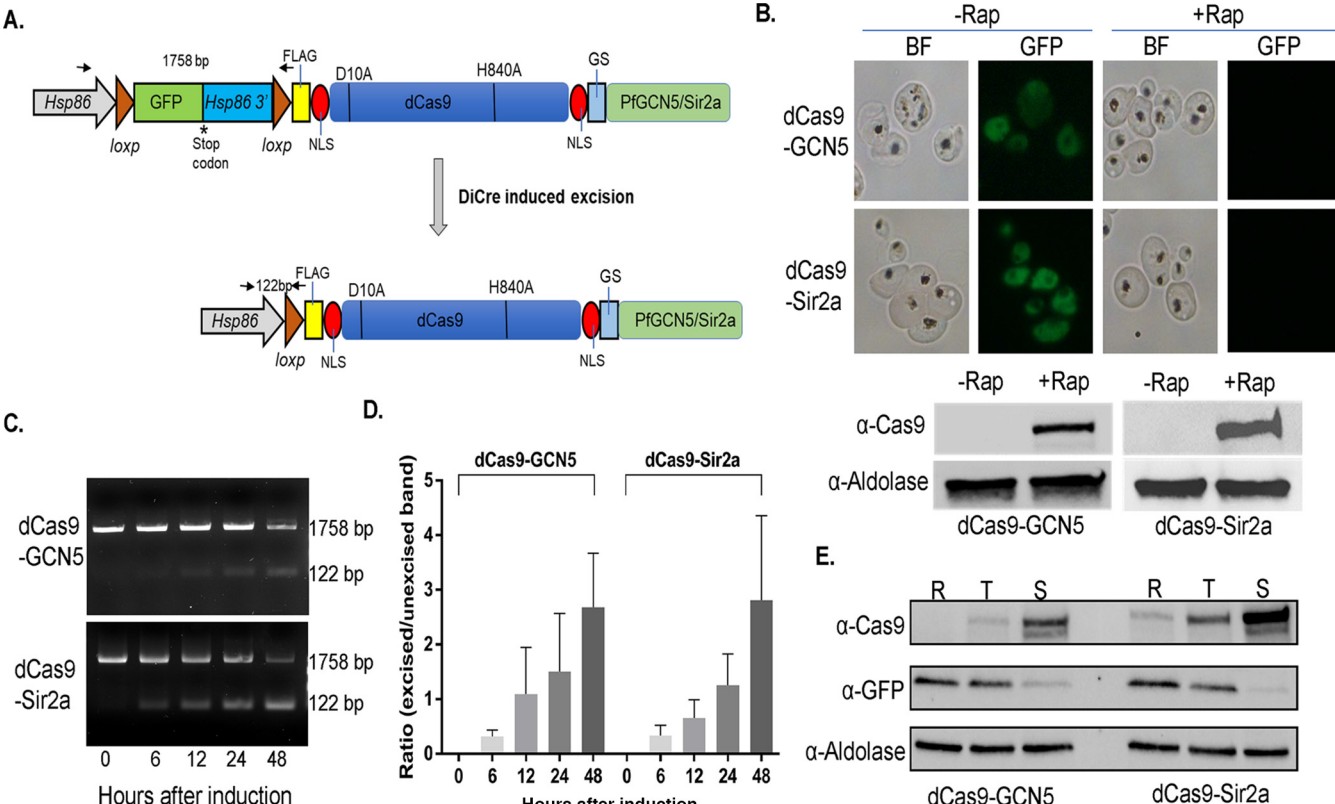

**FIG 1** Generation of inducible dCas9-GCN5/Sir2a systems using DiCre/*loxP*. (A) Schematic diagram illustrates the DiCre/*loxP*-inducible CRISPR/dCas9 system. GFP with stop codon and *Hsp86* 3'UTR flanked by *loxP* sequence was inserted before dCas9-GCN5/Sir2a. Rap is added to activate DiCre to remove sections between *loxP* sites, leading to the expression of dCas9-GCN5/Sir2a. FLAG, FLAG tag; D10A and H840A, two mutations in Cas9; NLS, nuclear localization signal; GS, glycine-serine protein domain linker. Two small black arrows in opposite directions show the locations of primers for PCR in panel C. The numbers between the arrows indicate the expected sizes of PCR products. (B) Live images and Western blots show the GFP and dCas9-fusion protein expression before (−Rap) and 48 h after (+Rap) DiCre was activated by the addition of Rap. (C) The efficiency of DiCre excision of *loxP*ed GFP cassette was measured by PCR with specific primers shown in panel A. The gradual decrease and increase of unexcised and excised bands at the sizes 1,758 bp and 122 bp are shown in agarose gels, respectively. (D) Three biological replicates were performed for the experiment in panel C to depict a bar graph showing the ratio of excised to unexcised bands over time elapsed after Rap treatment; pixel intensities of DNA bands were measured using ImageJ. BF, bright field. (E) Western blots show the induction of dCas9-GCN5 and dCas9-Sir2a protein expression and decrease of GFP expression at ring (R), trophozoite (T), and schizont (S) stages after Rap treatment. Rap was added at 0 to 6 h postinvasion (hpi) for 2 h.

selected, all except *PfDNMT* or *PfAP2-G* are essential for the IDC. For each gene, 2 to 3 sgRNAs were selected around the transcriptional start sites (TSSs) (Fig. 2B, 3A, 4A, Table S2) (32, 33). The constructs expressing these sgRNAs were transfected individually into either the *loxP*ed GFP-dCas9-Sir2a or the *loxP*ed GFP-dCas9-GCN5 parasite line. Resistant parasites after selection harboring different sgRNAs were treated with Rap at 0 to 6 h postinvasion (hpi) for 2 h. Total RNA was harvested at 40 to 46 hpi to measure the target gene mRNA level by reverse transcriptase-quantitative PCR (RT-qPCR).

Since the CRISPRi/a system developed here would inhibit or activate gene expression by modifying the chromatin environment near the TSS of the target genes, we were interested in using this system to downregulate highly expressed genes, regulate intermediately expressed genes in both directions, and activate lowly expressed or silent genes. For all genes tested, we could change the levels of target genes expression after Rap-induced dCas9-GCN5 or dCas9-Sir2 expression, but the extent of the change varied greatly among genes and was influenced substantially by the locations of the sgRNAs relative to the major TSS (Fig. 2C to H).

For the *PfGCN5* gene, the two sgRNAs selected were both embedded in the multiple TSSs that span ~1.2 kb in the 5' UTR of the gene, with gRNA1 located ~400 bp from the major TSS and gRNA2 much farther (~1 kb) from the major TSS (Fig. 2B). As expected, gRNA1 resulted in ~2-fold downregulation of PfGCN5 in the *loxP*ed GFP-dCas9-Sir2a parasite, whereas gRNA2 did not cause any changes in *PfGCN5* expression

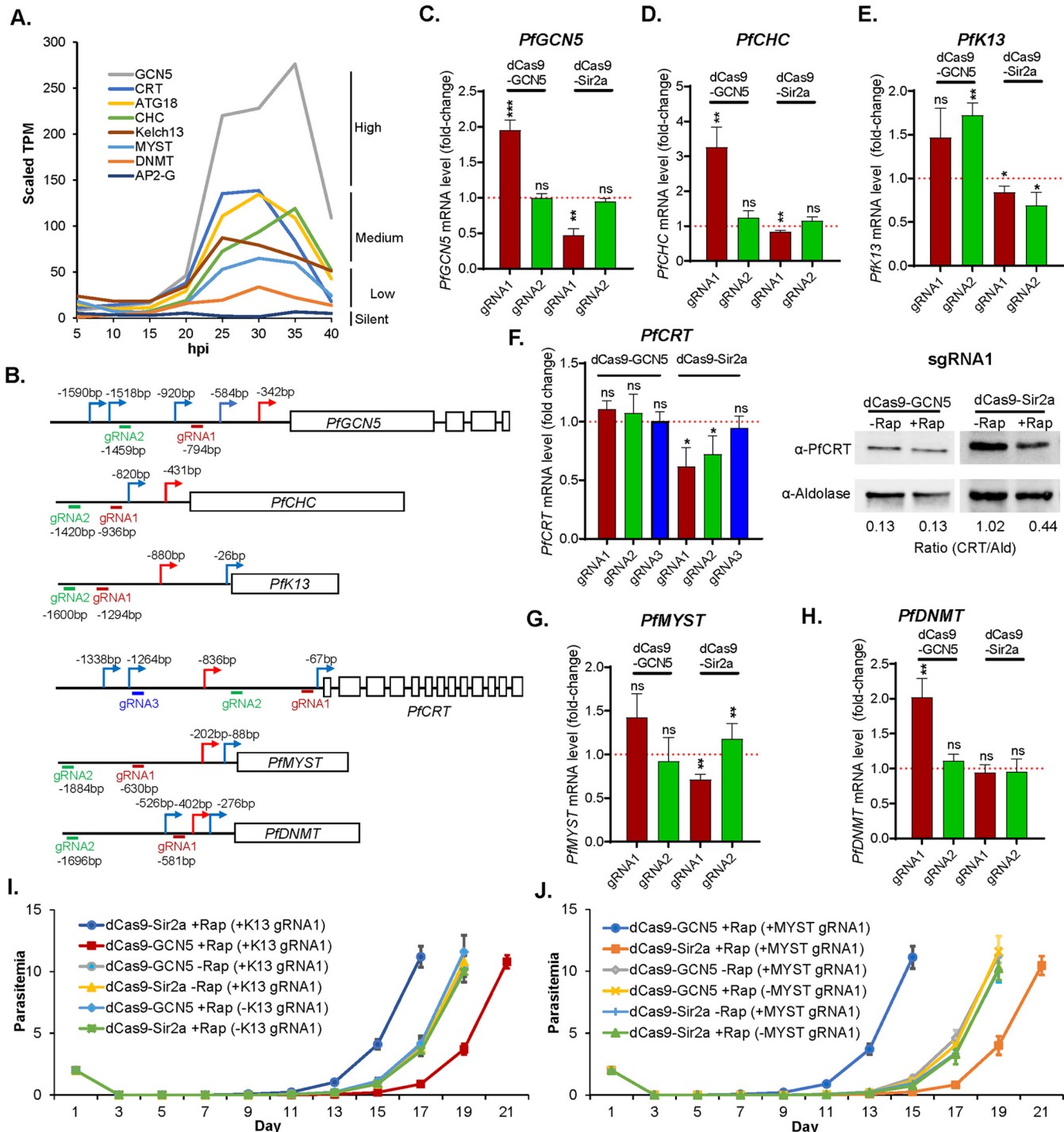

**FIG 2** Repression and activation of selected genes by inducible dCas9 systems. (A) Transcriptional expression levels of eight selected genes during IDC based on the transcriptomic profile by RNA-seq. (B) Schematic diagrams show the location of TSSs and gRNAs in six selected genes. (C to E) The transcriptional changes of *PfGCN5*, *PfCHC*, and *PfK13* in *loxPed* GFP-dCas9-GCN5 or Sir2a parasite lines after Rap treatment, respectively. (F) Left panel: RT-qPCR analysis of *PfCRT* transcript using 3 different sgRNAs (gRNA1 to 3) with dCas9-GCN5 and dCas9-Sir2a systems. Right panel: Western blot of PfCRT protein expression levels after activation and repression of *PFCRT* by using an anti-PfCRT antibody with anti-aldolase as a loading control. (G and H) The transcription levels of *PfMYST* and *PfDNMT* in the *loxPed* GFP-dCas9-GCN5 or Sir2a parasite lines after Rap treatment. Three replicates of RT-qPCR were performed for each gRNA and the paired *t* tests were conducted (ns, not significant; *, $P < 0.05$; **, $P < 0.01$; ***, $P < 0.001$). (I and J) Three replicates of recovery assays show the parasite growth after DHA treatment of parasite lines targeting *PfK13* (I) and *PfMYST* (J) by dCas9-GCN5/Sir2a with respective gRNA1, compared to the parasite lines with gRNA but without Rap induction and the parasite lines without gRNA but with Rap induction.

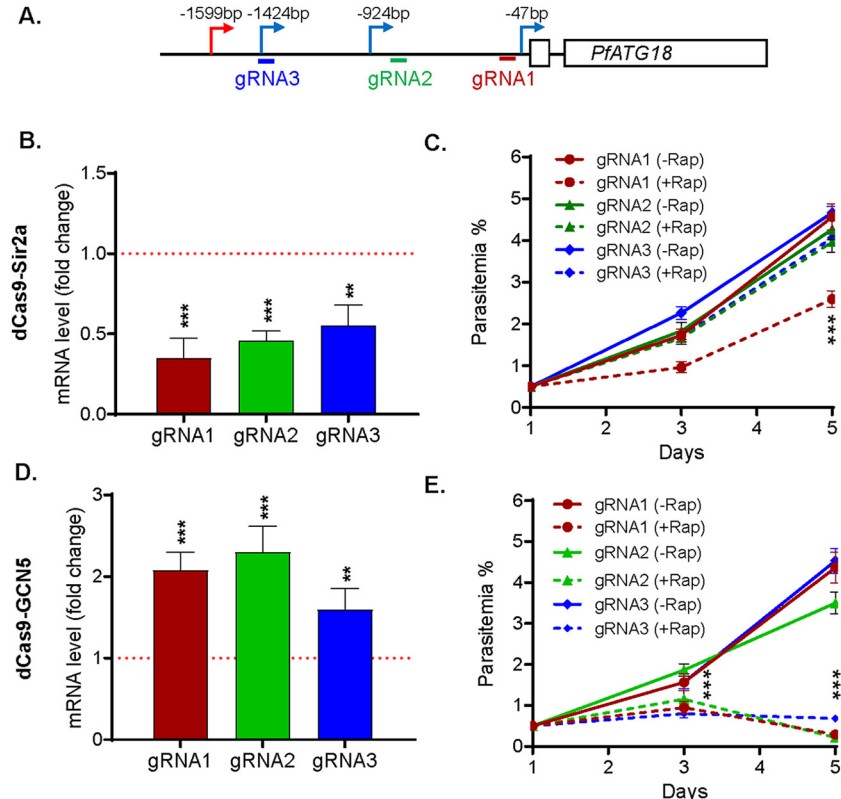

**FIG 3** Repression and activation of *PfATG18* by inducible dCas9 system. (A) Schematic diagram shows the location of TSSs and gRNAs in the 5′ UTR of *PfATG18*. (B and D) The changes of *PfATG18* transcriptions were analyzed by RT-qPCR in dCas9-Sir2a (B) and dCas9-GCN5 (D) parasite lines with gRNA1-3 after Rap treatment compared to the transcriptions without Rap treatment, respectively. Paired *t* tests were conducted (**, $P < 0.01$; ***, $P < 0.001$) from three replicates. (C and E) Parasite growth phenotype after repression and activation of *PfATG18* using gRNA1-3 in dCas9-Sir2a (C) and dCas9-GCN5 (E) systems was investigated by measuring the parasitemia from three replicates, respectively. Compared to the growth rate in the parasites without Rap treatment (−Rap), parasites harboring gRNA1 in dCas9-Sir2a system grew significantly more slowly at day 5 posttreatment (+Rap) (***, $P < 0.001$, two-way ANOVA), whereas parasites harboring any gRNA (1/2/3) in dCas9-GCN5 system grew significantly more slowly at days 3 and 5 posttreatment (+Rap) (***, $P < 0.001$, two-way ANOVA).

(Fig. 2C). Interestingly, despite that *PfGCN5* is a relatively highly expressed gene, gRNA1 in the *loxPed* GFP-dCas9-GCN5 parasite still increased *PfGCN5* expression by ∼2-fold (Fig. 2C). Although *PfGCN5* is an essential gene for the IDC, these changes in *PfGCN5* mRNA levels did not result in significant changes in asexual growth under the standard culture conditions. Similarly, gRNA1 and gRNA2 for *PfCHC* were located at 0.5 and 1 kb, respectively, upstream of the major TSS (Fig. 2B). Whereas gRNA1 led to an ∼3-fold increase in *PfCHC* expression in the *loxPed* GFP-dCas9-GCN5 parasite line, gRNA2 did not have any noticeable effect on *PfCHC* expression (Fig. 2D).

Two drug resistance genes, *PfK13* and *PfCRT*, were selected to evaluate both upregulation and downregulation of expression. The two gRNAs for PfK13 were both relatively far from the main TSS (∼400 bp and ∼700 bp, respectively) (Fig. 2B). In the *loxPed* GFP-dCas9-GCN5 parasite line, both gRNAs led to 1.4- to 1.7-fold upregulation of PfK13, whereas their downregulation effects in the *loxPed* GFP-dCas9-Sir2a parasite line were very modest (20 to 30% reduction) (Fig. 2E). For *PfCRT*, none of the three gRNAs had any evident effects on PfCRT expression in the *loxPed* GFP-dCas9-GCN5 parasite line, whereas gRNA1 and gRNA2, both located downstream of the major TSS, resulted in ∼40% and ∼30% reductions, respectively, in *PfCRT* mRNA level in the *loxPed* GFP-dCas9-Sir2a parasite line (Fig. 2F). We further compared the PfCRT protein levels in these parasite lines expressing gRNA1. We found that PfCRT protein was

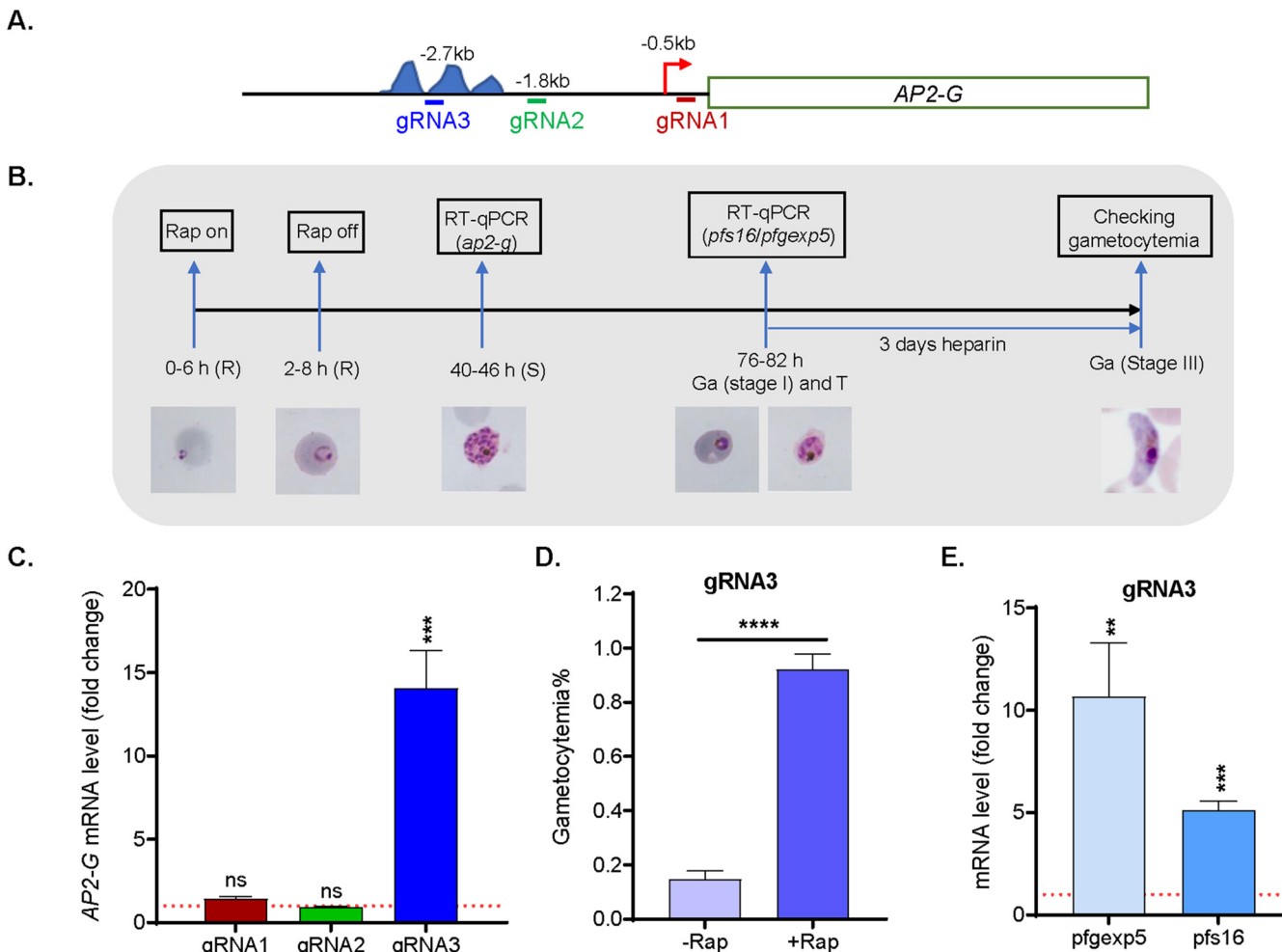

**FIG 4** Activation of *AP2-G* by the inducible dCas9-GCN5 system. (A) Schematic diagram shows the location of TSS and gRNAs in the 5′ UTR of *AP2-G*. The areas containing AP2-G/G5 binding motifs are labeled with three blue peaks. (B) A diagram illustrates the experimental workflow for the analysis of activation of *AP2-G* by the inducible system dCas9-GCN5. R, ring; T, trophozoite; S, schizont; Ga: gametocyte. (C) The changes of *AP2-G* transcription were analyzed by RT-qPCR in the parasites harboring inducible dCas9-GCN5 system and one of three gRNAs (gRNA1 to 3) compared to the transcription without Rap induction. Paired *t* tests were conducted (ns, not significant; ***, $P < 0.001$) from three replicates. (D) Three replicates of gametocytemia were counted 5 days after parasites harboring inducible dCas9-GCN5 system and gRNA3 were treated by Rap (+Rap) compared to the same parasites without Rap treatment (−Rap) (****, $P < 0.0001$, paired *t* test). (E) The two early gametocytes makers, *pfgexp5* and *pfs16*, were analyzed by RT-qPCR in the parasites harboring the dCas9-GCN5 system and gRNA3 at 76 to 82 hpi when the gametocytogenesis started (stage 1) and asexual parasites were at the trophozoite stage. The paired *t* tests were conducted (**, $P < 0.01$, ***, $P < 0.001$) from three replicates.

decreased by >50% in the *loxped* GFP-dCas9-Sir2a parasite line after Rap induction (Fig. 2F), consistent with the change in *PfCRT* mRNA level.

We were also interested in using the dCas9-GCN5 system to upregulate lowly expressed genes. For the two epigenetic regulators selected for evaluation, the two gRNAs for *PfMYST* were more than 400 bp upstream of the major TSS, while gRNA1 for *PfDNMT* was 180 bp from the major TSS (Fig. 2B). Regardless of the locations of the gRNAs, no noticeable or modest downregulation of *PfMYST* or *PfDNMT* was observed in the *loxPed* GFP-dCas9-Sir2a line. In the *loxPed* GFP-dCas9-GCN5 parasites, the *PfMYST* gRNA1 led to ~1.5-fold upregulation of *PfMYST* expression, while *PfDNMT* gRNA1 increased *PfDNMT* expression by almost 2-fold (Fig. 2G and H).

We monitored the growth phenotypes of the parasites only after we induced changes in the mRNA levels of the tested genes. Except for *PfATG18* and *PfAP2-G*, no changes in the growth curve and the developmental progression of the parasite lines were observed when the target genes were regulated using the dCas9-GCN5 and the dCas9-Sir2a approaches.

**Altered sensitivities to artemisinin after regulation of *PfK13* and *PfMYST*.** *PfK13* mutation is known as the marker of artemisinin resistance (34, 35), while *PfMYST* was found to be involved in DNA repair (36). Recently, enhanced DNA repair was also identified to be associated with artemisinin resistance (37). To reveal whether alteration of expression in *PfK13* and *PfMYST* could change the sensitivities to artemisinin, we treated *loxPed* GFP-dCas9-GCN5/Sir2a parasite lines harboring gRNA1 targeting *PfK13* and *PfMYST* with 1 $\mu$M dihydroartemisinin (DHA) at the early ring stage for 24 h and measured the time of parasite reappearance (recovery assay). Compared to the parasite lines with gRNA but without Rap induction and the parasite lines without gRNA but with Rap induction, downregulation of *PfK13* (dCas9-Sir2A-K13gRNA1, +Rap) and upregulation of *PfMYST* (dCas9-GCN5-MYSTgRNA1, +Rap) resulted in earlier recovery, whereas upregulation of *PfK13* (dCas9-GCN5-K13gRNA1, +Rap) and downregulation of *PfMYST* (dCas9-Sir2A-MYSTgRNA1, +Rap) led to later recovery (Fig. 2I and J). These results are consistent with the finding that the decreased K13 abundance reduced the artemisinin resistance (38–40) and suggest that *PfMYST* is involved in artemisinin resistance probably by the protection of DNA damage caused by artemisinin (41).

**Growth inhibition resulting from bidirectional regulation of *PfATG18* expression.** *PfATG18* is essential for the IDC (42–44), and a mutation in *PfATG18* was associated with decreased sensitivities to DHA, artemether, and piperaquine in the samples from the China-Myanmar border (45). The major TSS of *PfATG18* is 1.6 kb from the ATG, and the three gRNAs selected are all located downstream from the major TSS. Interestingly, regardless of the locations of the gRNAs, all led to an $\sim$2-fold reduction in *PfATG18* expression in the *loxPed* GFP-dCas9-Sir2a parasite lines. Specifically, the *PfATG18* mRNA levels were decreased to $\sim$30%, 50%, and 60% of those in untreated parasites for gRNA 1, 2, and 3, respectively (Fig. 3B). Only the parasite line with gRNA1 yielding $\sim$70% transcript reduction showed growth retardation, with the day 5 parasitemia reaching 2.6% compared to 4.7% in the untreated parasites (Fig. 3B and C; analysis of variance [ANOVA], $P < 0.001$). In contrast, the other two parasite lines did not show noticeable growth defects, suggesting that the degree of *PfATG18* mRNA reduction at 50% or lower did not result in a growth phenotype change under the standard culture conditions (Fig. 3C). It is noteworthy that *PfATG18* knockdown (KD) using the destabilization domain also resulted in a slower growth phenotype (42).

Early studies found that episomal expression of *PfATG18* did not cause any noticeable changes in growth phenotype (44, 46). Since it is not known whether episomal expression resulted in *PfATG18* overexpression, we wanted to see whether conditional upregulation of *PfATG18* in the *loxPed* GFP-dCas9-GCN5 parasite line would lead to growth phenotype changes. After Rap treatment at the ring stage (0 to 6 hpi), the *PfATG18* mRNA measured at 40 to 44 hpi was elevated by $\sim$2-, 2.3-, and 1.6-fold in parasites expressing gRNA1, 2, and 3, respectively, compared to that of the untreated parasites (Fig. 3D). Surprisingly, all parasite lines showed severe growth retardation on day 3 and day 5, with more profound effects starting from the second IDC (Fig. 3E, ANOVA, $P < 0.001$).

**Activation of HP1-controlled silent gene *PfAP2-G* via dCas9-GCN5.** In *P. falciparum*, heterochromatin protein 1 (HP1) binds H3K9me3 to control $\sim$425 genes localized mainly in the subtelomere regions (47–50). These include silent genes (e.g., *PfAP2-G*) and genes that are expressed in a mutually exclusive manner (e.g., the *var* gene family with $\sim$60 members) during the IDC. The ApiAP2 (AP2) transcription factor PfAP2-G is a master regulator of gametocytogenesis (51–54), and its activation led to the conversion of asexual-stage parasites to sexual development (55, 56). We selected *PfAP2-G* to evaluate whether the inducible dCas9-GCN5 system could activate the HP1-controlled silent gene during the IDC. Three gRNAs were selected. gRNA 1 located in the proximity of the weak TSS (−0.5 kb) for the lowly expressed *PfAP2-G* (17, 32, 33, 57) (Fig. 4A). Since PfAP2-G and PfAP2-G5 were found to regulate *PfAP2-G* expression by binding their respective motifs located between −2.0 kb and −3.5 kb (52, 58), gRNAs 2 and 3 were designed to localize downstream and within (−1.8 kb and −2.7 kb) these motifs, respectively (Fig. 4A). In the *loxPed* GFP-dCas9-GCN5 parasite line, the mRNA

level of *PfAP2-G* was measured at 40 to 46 hpi after the dCas9-GCN5 expression was induced at the ring stage (0 to 6 hpi) (Fig. 4B and C). RT-qPCR analysis of *PfAP2-G* did not detect any increased expression of *PfAP2-G* in parasites with either gRNA1 or 2 over the uninduced controls (Fig. 4C), despite that the gRNA1 is localized close to the TSS of the gene. In contrast, *PfAP2-G* expression was increased by ~14-fold in the parasite line harboring gRNA3 (Fig. 4C).

To determine whether elevated *PfAP2-G* expression was correlated with increased gametocytogenesis, parasite cultures were treated with heparin for 3 days starting from the second cycle after Rap treatment (76 to 82 hpi) to eliminate asexual-stage parasites (Fig. 4B). The *loxPed* GFP-dCas9-GCN5 line harboring gRNA3 had an increase in gametocytemia more than 4-fold that in the untreated parasites (Fig. 4D, unpaired two-tailed Student's test, $P < 0.0001$). As expected from the mRNA analysis, the other two gRNAs did not lead to an increase in gametocytemia (data not shown). There are many known early gametocyte markers, including *Pfs16* and *Pfgexp5* (52, 59, 60). Consistently, RT-qPCR analysis of parasites in the second cycle after Rap treatment (76 to 82 hpi) (Fig. 4B) showed concomitant upregulation of *Pfgexp5* and *Pfs16* by more than 10- and 5-fold, respectively, in the parasite line with gRNA3 compared to that in the untreated parasites (Fig. 4E). These results demonstrated that the silent gene *PfAP2-G* could be conditionally activated in the inducible dCas9-GCN5 system, leading to elevated sexual conversion.

**The inducible CRIPRi/a systems regulate gene expression in gametocytes.** The DiCre/loxP system has been successfully used to excise *loxPed* genes in gametocytes (61). This triggered us to determine whether CRIPRi/a systems can regulate gene expression in the gametocyte stage. Consistently, Rap treatment at day 5 (stage III) gametocytes induced excision of *loxPed* GFP cassette (Fig. S3A). *PfATG18* gRNA1 was chosen because it substantially altered the *PfATG18* expression in the asexual stage (Fig. 3). Forty-eight hours after Rap induction starting at stage III gametocytes, *PfATG18* mRNA was elevated by ~3.3-fold and decreased to ~30% in dCas9-GCN5 and -Sir2a parasite lines, respectively, compared to that in the Rap-induced parasite line without gRNA1 or noninduced gametocyte with gRNA1 (Fig. S3B). Only upregulation of *PfATG18* expression resulted in a significant decrease of gametocytemia as measured 3 days after induction (Fig. S3C). Taken together, the inducible CRIPRi/a system functions in sexual-stage parasites.

## DISCUSSION

Our recent study has shown that epigenetic editing by dCas9 fused to the HAT domain of PfGCN5 or the deacetylase domain of PfSir2a allows for activating or silencing the gene of interest without leaving a genetic scar (28). This CRISPR/dCas9 system relies on the constantly coexpressed dCas9 and sgRNA, leading to strong activation/repression of the target genes. This constitutive expression could limit certain applications if the level of regulation of the target gene achieved with this dCas9 system is detrimental to the IDC. Therefore, we aimed to introduce inducibility to this dCas9-GCN5/Sir2a system. From experimentation with the TetR-DOZI and DiCre/*loxP* conditional modules to control the timing of CRISPR/dCas9 expression, we selected the DiCre/*loxP* module since there was no background expression of dCas9-GCN5 and -Sir2a before the addition of Rap, whereas TetR-DOZI conditional system showed a leaky expression (Fig. S1), and induction of dCas9 fusion proteins by addition of aTc did not result in more sex conversion (gametocyte) when *PfAP2-G* gRNAs were used (data not shown), probably because the leaky expression caused less magnification of induction than that of DiCre/*loxP* system. We then tested the suitability of this inducible system for regulating genes with expression levels ranging from highly expressed to silent genes during the IDC.

Our data further confirmed that the degree of gene activation or repression achieved with the inducible dCas9 systems depended primarily on the location of gRNA relative to the TSSs of the target genes. In most cases, the sgRNAs located closest

to the major TSSs produced the most potent activation or repression effect (Fig. 2). Additionally, from the results of *PfCRT* and *PfATG18*, gRNAs located downstream of major TSSs lead to the repression of target genes, probably due to the blockade of the transcriptional process by the gRNA/dCas9 complex. Since most *P. falciparum* genes use multiple TSSs, selecting several sgRNAs around the TSSs will identify the one with the strongest effects on the target gene. For *PfAP2-G*, only the sgRNA3 localized to the PfAP2-G/PfAP2-G5 binding motifs triggered substantial activation of *PfAP2-G* expression (Fig. 4), suggesting the influence of the local chromatin status of the sgRNA-target site. Thus, additional factors such as transcription factor binding sites, the location of active or silent histone marks, and more accurate TSSs will need to be considered to optimize sgRNA selection. However, searching for suitable sgRNAs for Cas9 requiring the PAM sequence NGG in the extremely AT-rich genome of *P. falciparum* is challenging. More recently, a new type of Cas, Cas12a (also called Cpf1), has emerged as an alternative nuclease to Cas9 (62, 63). One advantage of Cas12a is that the preferred PAM sequence is T-rich (TTTV, V can be C, G, or A), which could alleviate the difficulties of designing sgRNAs in *P. falciparum*.

The inducible dCas9 system reported here offers *in situ* upregulation of genes to study the effect of gene overexpression. One approach to studying gene function is the overexpression of the full-length or truncated gene, often from an episome. However, continuous overexpression might cause the parasite to adapt to the change such as the attenuation of the endogenous target gene, which may mask the true phenotype resulting from overexpression of the target gene. Continuous episomal overexpression of a target gene sometimes is toxic or detrimental to the parasite. For example, ectopic overexpression of PfSR1 was toxic and caused the parasites to constrain PfSR1 overexpression for survival (64). In *Toxoplasma gondii*, transient but not stable overexpression of MYST-A was achieved (65). Here, we showed that the inducible dCa9-GCN5 system allowed upregulation of almost all genes tested, even including highly expressed genes such as *PfGCN5* and *PfCHC*. Interestingly, overexpression of *PfATG18* by ~2-folds in the dCas9-GCN5 parasites resulted in substantial growth defects. In contrast, no growth defect was reported when *PfATG18* was episomally expressed (44, 46), suggesting that the parasites might have adapted to the constant overexpression by attenuating endogenous gene expression. Although we did not test the potential off-target effects of the sgRNAs, it is unlikely that all three strictly selected sgRNAs for *PfATG18* had additional targets in the *P. falciparum* genome. Similar growth defects were also found when *PfATG18* transcription was increased by more than 3-fold in gametocytes, suggesting that *PfATG18* is critical for the parasite survival given that this gene was found to be involved in the food vacuole dynamics, autophagy-like pathway, and apicoplast biogenesis in the parasites (42, 44, 46).

We have shown that the activation of the DiCre recombinase by Rap was rapid, resulting in high-level induction of the dCas9-fusion proteins in the same IDC (Fig. 1). With this, observation of phenotypic changes may begin in the first IDC or, at most, the second IDC, which was illustrated for the *PfATG18* gene (Fig. 3). Although under the standard culture conditions no growth phenotype changes were observed for six of the eight genes tested, the phenotypic changes under other conditions (e.g., stress, drug exposure) probably will elucidate the tested genes' functions. Indeed, alteration of the expression of *PfK13* and *PfMYST* resulted in changes in sensitivities to artemisinin (Fig. 2). Although dCas9-GCN5 activated *PfAP2-G* and displayed a significantly higher sexual conversion (~1%), this conversion rate is relatively limited compared to the extremely high rate (~90%) when *PfAP2-G* was driven by a strong constitutive promoter (calmodulin) (55).

Collectively, we showed the values of the DiCre/*loxP*-based, inducible dCas9-GCN5/Sir2a systems for functional gene studies in *P. falciparum*. To increase the robustness of the gene regulation, multiple sgRNAs should be tried to select the one with the strongest effect and no off-targets. In addition, replacing the dCas9 with dCas12 in this system may alleviate the challenges in choosing the most appropriate sgRNAs.

## MATERIALS AND METHODS

**Molecular cloning.** To generate TetR-based inducibility into the dCas9-GCN5 and -Sir2a systems, the 10× aptamer was amplified from the pMG75 ATPase4 plasmid (8, 9) using the primers aptamer F/R (Table S3) and inserted into 3′ UTR of the dCas9-GCN5/Sir2a expression cassette at PacI, resulting in pdCas9-GCN5/Sir2a-10× aptamer. We generated a TetR-DOZI-expressing plasmid, pTetR-DOZI, from pMG75 ATPase4 by deleting the ATPase4 homologous region and the 10× aptamer-containing 3′ UTR.

To integrate the DiCre/*loxP* conditional module into the dCas9-GCN5 or dCas9-Sir2a expression cassette (28), we amplified *gfp* flanked by two *loxP* sequences using primers LoxP-GFP-F and LoxP-GFP-R (Table S3). The PCR product was inserted into XhoI-digested vector pUF1-dCas9-GCN5 or -Sir2a using the in-fusion HD cloning kit (Clontech). The *hsp86* 3′ UTR was amplified using primers Hsp86-F and Hsp86-R (Table S3) and cloned at the XmaI site in pUF1-dCas9-GCN5 or -Sir2a, immediately downstream of *gfp*. Two or three sgRNAs were designed for each tested gene based on predicted TSS sites according to the sgRNA selection criteria (Table S2 and S3) (17). Individual sgRNAs were cloned into the plasmid pL6-CS-WR (28).

**Plasmodium culture and transfection.** The *P. falciparum* 3D7 clone was used for generating the TetR-based expression system. A *P. falciparum* clone engineered to express DiCre constitutively (5) was kindly provided by Michael Blackman. Parasites were maintained in human red blood cells (RBCs) and cultured at 37°C in a gas mixture of 5% $CO_2$, 3% $O_2$, and 92% $N_2$ as described previously (36). Parasite transfection was done using the RBC loading method (66). For the TetR-based inducible dCas9-GCN5 or -Sir2a system, pTetR-DOZI and pdCas9-GCN5/Sir2a-10× aptamer plasmids were transfected into 3D7, followed by drug selection with WR99210 at 2.5 nM and blasticidin-S-HCl (BSD) at 2.5 $\mu$g/mL. The DiCre/*loxP*-based plasmids were introduced into the DiCre-expressing parasite line, and resistant parasites were selected with BSD.

**Rapamycin-induced excision and dCas9-fusion protein expression in the DiCre/*loxP* system.** Highly synchronized rings were obtained by incubating purifying schizonts via 65/35% Percoll gradient with fresh RBCs for 4 h. Rap at 100 nM was added to the synchronized rings for 2 h, and parasites were harvested 6, 12, 24, and 48 h later. The same experiments were also performed starting from the trophozoite and schizont stages. For excision analysis in gametocytes, the synchronized gametocytes were induced by previously described methods (67). On day 2 after gametocyte induction, heparin was added to the culture for 3 days to eliminate asexual-stage parasites. Rap at 100 nM was added to day 5 (stage III) gametocytes for 2 h, and the gametocytes were harvested 6, 12, 24, and 48 h later. Parasite genomic DNA was isolated using the proteinase K digestion and phenol-chloroform extraction method. PCR was performed with the same amount of genomic DNA using primers Ex-F and Ex-R (Table S3). PCR products were separated by agarose gel and stained with ethidium bromide. Pixel intensities of DNA bands were measured using ImageJ. Three biological replicates were performed for each experiment.

To measure the expression of the dCas9 fusion proteins and GFP in the parasites before and after DiCre excision events, rings were treated with Rap for 2 h and then were harvested 6, 26, and 40 h later to represent ring, trophozoite, and schizont stages, respectively. Western blotting was done with mouse anti-GFP (1:3,000, ThermoFisher), anti-Cas9 (1:1,000, Sigma), and rabbit anti-PfAldolase (1:3,000, kindly provided by Tobias Spielmann) antibodies as primary antibodies, followed by horseradish peroxidase (HRP)-conjugated goat anti-rabbit IgG and goat anti-mouse IgG (Sigma) as secondary antibodies. The GFP fluorescence signals in live parasites were captured using a Nikon Eclipse E600 epifluorescence microscope.

**Induced dCas9-fusion protein expression in the TetR-DOZI system.** Western blotting was used to measure dCas9 expression in the TetR-DOZI-based inducible dCas9-GCN5 or -Sir2a system. Ring-stage parasites expressing both TetR-DOZI and the dCas9-fusion proteins were equally divided into two cultures and were treated with aTc at 0.5 $\mu$M or ethanol (vehicle control) for 30 h. Then, parasites were harvested for Western blotting with rabbit anti-FLAG antibodies (1:1,000, Abcam) and rabbit anti-PfAldolase as the primary antibodies and HRP-conjugated goat anti-rabbit IgG as secondary antibodies. The wild-type 3D7 was used as a negative control.

IFA was performed to detect the expression of dCas9 according to an established method (68, 69). Briefly, parasitized RBCs were fixed with 4% paraformaldehyde and 0.0075% glutaraldehyde, followed by quenching with 50 mM glycine. Fixed cells were then treated with 0.5% Triton X-100 and blocked in 3% bovine serum albumin (BSA) for 1 h. The anti-FLAG antibodies and fluorescein isothiocyanate (FITC)-conjugated goat anti-rabbit IgG (Sigma) were used as primary and secondary antibodies. Images were captured using a Nikon Eclipse E600 epifluorescence microscope.

**RNA extraction and RT-qPCR.** Parasites were isolated after treating the parasitized RBCs with 0.1% (vol/vol) saponin, and total RNAs were extracted using the Quick-RNA MiniPrep kit (Zymo Research). Reverse transcriptase and real-time PCR were performed by using SuperScript III RT (Invitrogen) and Faststart Universal SYBR green master mix (Roche). The relative expression of each gene in the Rap- or dimethyl sulfoxide (DMSO)-treated parasites was normalized to seryl-tRNA synthetase (PF3D7_0717700) and calculated using the $2^{-\Delta\Delta CT}$ method (70). All the qPCR primer sequences are listed in Table S3. Three biological replicates were performed for each experiment.

**Growth phenotypic analysis.** The growth phenotype of *Pfatg18* gRNA1-3 parasite lines was compared in triplicate. Cultures were tightly synchronized as described above. Cultures starting at 0.5% ring were treated with Rap or DMSO, and parasitemia was monitored daily using Giemsa staining. Phenotypic analysis of *Pfap2-g* was performed following the experimental workflow in Fig. 3. Briefly, the highly synchronized ring-stage parasites (0 to 6 hpi) were immediately treated with Rap or DMSO for 2 h. The transcription of *Pfap2-g* and early gametocyte markers (*pfgexp5/pfs16*) was measured at 40 to 46 hpi and 76 to 82 hpi using RT-qPCR (Fig. 3B, Table S3), respectively. Heparin was added afterward for 3 days to eliminate asexual-stage parasites (71). The gametocytemia was calculated after heparin

treatment using Giemsa-stained smears. To study the gametocyte growth phenotype after targeting *PfATG18*, the synchronized gametocytes were induced by the methods described previously (67). On day 2 after gametocyte induction, heparin was added to the culture for 3 days to eliminate asexual-stage parasites. Rap at 100 nM was added to day 5 gametocytes for 2 h and the gametocytemia was measured on the next day by Giemsa staining for 4 days.

**Recovery assay.** This assay was performed based on established methods with some modifications (72–75). Briefly, the highly synchronized early ring stage (1 to 3 h) parasites at 2% parasitemia were treated with 1 $\mu$M DHA for 24 h. Subsequently, the parasites were cultured in a drug-free medium and parasite growth was examined daily using Giemsa staining. The amount of time needed for each parasite culture to reach 10% parasitemia was recorded.

**Statistical analysis.** For growth phenotype experiments, three independent biological replicates were performed. The results are presented as mean $\pm$ standard deviation (SD) and analyzed by one- or two-way ANOVA or paired *t* test. For gene expression (RT-qPCR) upon induction by Rap and DSMO, $\Delta CT$s from +Rap and −Rap from three biological replicates were analyzed by paired *t* test.

## SUPPLEMENTAL MATERIAL

Supplemental material is available online only.
**SUPPLEMENTAL FILE 1**, PDF file, 0.6 MB.

## ACKNOWLEDGMENTS

This study was funded by the National Institute of Allergy and Infectious Diseases, National Institutes of Health (1R21AI149202 to J.M.), and the startup fund from Morsani College of Medicine to J.M. We want to thank Michael Blackman for providing a *P. falciparum* clone engineered to express DiCre constitutively and Tobias Spielmann for providing the anti-PfAldolase antibodies.

J.M. and L.C. conceptualized the study. B.X. and L.J. provided resources. X. Liang, R.B., and F.A.S. developed the methodology and performed the investigations. X. Li, J.Q., B.X., and H.M. supported the investigation. J.M., L.C., and L.J. reviewed and edited the paper, and J.M. supervised the study.

We declare no conflicts of interest.

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
