## [Reviewer comments · Microbiology Spectrum]

Microbiology Spectrum

A leak-free inducible CRISPRi/a system for gene functional studies in *Plasmodium falciparum*

Xiaoying Liang, Rachasak Boonhok, Faiza Siddiqui, Bo Xiao, Xiaolian Li, Junling Qin, Hui Min, Lu-bin Jiang, Liwang Cui, and Jun Miao

Corresponding Author(s): Jun Miao, University of South Florida

Review Timeline:

Submission Date:	December 30, 2021
Editorial Decision:	February 14, 2022
Revision Received:	April 15, 2022
Accepted:	April 18, 2022

Editor: Björn Kafsack

Reviewer(s): Disclosure of reviewer identity is with reference to reviewer comments included in decision letter(s). The following individuals involved in review of your submission have agreed to reveal their identity: Heather J. Painter (Reviewer #2)

Transaction Report:

DOI: <https://doi.org/10.1128/spectrum.02782-21>

February 14, 2022

Dr. Jun Miao
University of South Florida
Tampa

Re: Spectrum02782-21 (A leak-free inducible CRISPRi/a system for gene functional studies in *Plasmodium falciparum*)

Dear Jun,

Both the reviewer felt that the manuscript and new regulatory system would be a valuable addition to the field but had some concerns that need to be addressed.

Link Not Available

Sincerely,

Björn Kafsack

Journals Department
Reviewer comments:

Reviewer #1 (Comments for the Author):

Conditional knockdown systems in *P. falciparum* have accelerated the study of this deadly human parasite. One major bottleneck remains the need to clone AT-rich homology regions that are used to repair Cas9-derived injuries. Unlike other eukaryotic organisms, *Plasmodium* does not possess RNAi-like systems where the expression of a short RNA results in specific gene knockdown. Previous work, including from the authors of this manuscript, used catalytically-dead Cas9 (dCas9) to knockdown gene transcription via the expression of a small guide-RNA targeting specific genes. The authors have previously demonstrated the utility of using transcriptional activators and repressors to enhance the effect to dCas9 on gene expression. In these studies, dCas9 was constitutively expressed and therefore, its utility was limited for the study of essential genes for

asexual growth in vitro.

In the current, manuscript, Liang et al build on their previous work and use the DiCre recombinase system for conditional expression of dCas9 fused to transcriptional activators and repressors. This is an advance for the field because this CRISPRi/a system can now be used to study the function of genes essential for asexual growth. Furthermore, these tools could one day allow high-throughput screening efforts to investigate *P. falciparum* biology.

The effect of CRISPRi/a was then tested on several genes required for asexual growth, gametocytogenesis, and a gene with a putative role in drug resistance. The data are mostly well controlled and the authors should be commended for their comprehensive approach. The manuscript is well written and the conclusions are supported by the data.

Fig. 1C and D: How many times was this experiment performed? There are no statistics for the graph.

Fig. 2B and D: No statistics are provided for the bar graphs. This is important to understand the relationship between gene knockdown and the observed phenotype.

Fig. 3C and E: The statistics are missing again and it is not clear how many times the experiment was repeated. It would be useful to the reader to have this information in the figure legends.

Fig. S2: The bar graphs have not been statistically analyzed. The number of biological repeats are not provided.

Lines 176-181: There is no figure referenced for these results.

Reviewer #2 (Comments for the Author):

In the submitted manuscript, Liang et al. sought to develop a conditional CRISPR1/a system in the malaria parasite, *P. falciparum*, that allowed for a more controlled expression of dCas9. Thus, enabling the more accurate study of the effects of gene expression induction or knockdown to the parasite's growth and biology. The authors develop a DiCre recombinase inducible excision within *P. falciparum* to regulate the expression of either dCas9-GCN5 or -Sir2a. It is unclear whether the authors have achieved a fully controllable system and, while the level of gene expression changes, this system does not always produce the desired impact. While this study provides a new tool in the molecular arsenal for understanding malaria parasite biology, this system is still plagued by the fact that the results may still be influenced by the leaky expression of dCas9 and the resulting moderate levels of induction/repression of gene of interest. However, it will be useful when applied in conjunction with other tools already available.

Major Comments:

1. While the authors sought to create a non-leaky, leaky inducible system, they fail to address the low level of excision that occurs at 0h after induction (Figure 1D) and the lack of protein expression of dCas9-GCN5 (Figure 1E). Have the authors examined the stage specific effects of the excision and expression of dCas9 carefully throughout the IDC in replicate? Does induction of excision always take 48 hours to be effective or is the result presented just a stage-specific effect?
2. It is curious that the authors chose to place much of the data that evaluates the efficacy of their system in the Supplement. It would be preferable that the authors move the data in Figure S2 to the main text. While the results can be confusing and not as definitive one would want in a new tool, this is the reality of the system. The results presented here are more impactful than the current Figure 2.
3. The authors use the "extreme" example of a stage-specific gene, ap2-g, whose expression drives the conversion of the malaria blood stage parasite from asexual to sexual development. While it was interesting to see the various levels of impact each guide location had on the induction of expression via dCas9-GCN5, the level of induction doesn't compare to current systems that exist. Additionally, the use of gexp5 as a "marker" of gametocyte gene upregulation is questionable because it is known to be expressed independent of AP2-G.

Minor Comments:

1. The authors state that there is no evidence of 'leaky' expression of dCas9-GCB5 or Sir2A in their system. This claim should be supported by evidence because the authors never directly compare the TetR-DOZI to the DiCre systems they have generated. The ratio of unexcised to excised in Fig 1D at time 0h suggests that the system may be a bit leaky. If so, is this detrimental to the parasite (with or without gRNAs, with or without RAP/aTc). Please include the Western Blot evidence in the supplemental and any additional evidence that the transgenic dCas9 parasite lines have the same/similar growth phenotype to wild type.
2. Could the authors comment further on why the TetR-DOZI system was abandoned? This induction may provide a functional tool for reversing any phenotypes from gene activation/repression.
3. In Figure 1A, the authors illustrate the DiCre/loxP inducible system. For clarity, what does the blue box just following the second NLS (prior to PfGCN5/Sir2a) represent? Why did the authors choose to include dual nuclear localization signals?
4. Could the authors comment as to why the excision in the GCN5 line is so much less efficient than in the Sir2a line?
5. The authors should clarify in their figure legends the number of biological/technical replicates and what the error bars represent. It is unclear how many times some of these experiments were replicated.
 - a. Please clarify if the excision ratio in Figure 1D was performed more than once. If not, provide a biological n >2 for the excision ratio.
 - b. Please clarify the number of replicates in Figure S2C-F. Again, a biological n > 2 is recommended.

- c. Lastly, please clarify the biological replicates of the Atg18 and AP2-G CRISPRi/a experiments.
6. To verify the expression of individual genes of interest, the authors extract RNA at 40-46hpi and compare +/-RAP parasites.
- Why did the authors choose 40-46hpi when the most efficient excision occurs later than that according to Fig. 1D? Did the authors examine the levels of mRNA during the next cycle following RAP treatment?
 - I am curious about the effects of RAP treatment on the parasite progression. Have the authors compared developmental progression of the DiCre parasites in the presence of RAP with and without the addition of gRNAs. Have the authors determined the changes in expression measured by a single timepoint RT-PCR are not purely due to any developmental delays from off target dCas9 interactions?
7. Could the authors speculate why does asexual replication decrease in both Sir2a/GCN5 ATG18 line? Do the authors perform these growth assays in biological triplicate for each guide or just triplicate in total (once for each guide)? Is there a growth phenotype with +/-RAP and no guides?
8. While the authors provide evidence for expression changes (RT-PCR) in both the Sir2A/GCN5 lines, there are no statistics provided (Figure S2) and there is a lack of any growth phenotype (noted in text, but no evidence provided) for "essential" genes that were targeted. Can the authors comment whether this system will be useful considering these results?
9. The authors state that, "since this study aimed to build inducibility into the dCas9 system, we did not vigorously evaluate the phenotypic changes under other conditions." While I agree that the goal was to build an inducible system, isn't the utility of that system linked to the ability to result in a phenotype? As such, two recommendations that could potentially strengthen this study are to:
- Correlate an altered expression of atg18 or k13 to drug sensitivity
 - Induce gene expression changes of a gametocyte-specific gene while evaluating the ability of RAP to induce recombinase activity in sexually developing parasites.
10. There are a few clarifications that are necessary for the "Materials and Methods":
- Distinguish which strain of *P. falciparum* that is being used for the study and describe its ability to make mature gametocytes which are transmission-competent.
 - Please confirm what stage the parasites are at 48h following RAP addition. This will establish whether the highest effects are in the same or next cycle.
 - Please describe the numbers of replicates and controls for each experiment.
 - Describe the statistical analyses that were performed to generate p-values in some experiments.

Staff Comments:

Preparing Revision Guidelines

Please return the manuscript within 60 days; if you cannot complete the modification within this time period, please contact me. If you do not wish to modify the manuscript and prefer to submit it to another journal, please notify me of your decision immediately so that the manuscript may be formally withdrawn from consideration by Microbiology Spectrum.

Responses to Reviewers' Comments

We thank the reviewers for your thorough review of our manuscript, your support, and constructive comments.

Reviewer #1

Conditional knockdown systems in *P. falciparum* have accelerated the study of this deadly human parasite. One major bottleneck remains the need to clone AT-rich homology regions that are used to repair Cas9-derived injuries. Unlike other eukaryotic organisms, *Plasmodium* does not possess RNAi-like systems where the expression of a short RNA results in specific gene knockdown. Previous work, including from the authors of this manuscript, used catalytically-dead Cas9 (dCas9) to knockdown gene transcription via the expression of a small guide-RNA targeting specific genes. The authors have previously demonstrated the utility of using transcriptional activators and repressors to enhance the effect to dCas9 on gene expression. In these studies, dCas9 was constitutively expressed and therefore, its utility was limited for the study of essential genes for asexual growth *in vitro*.

In the current, manuscript, Liang et al build on their previous work and use the DiCre recombinase system for conditional expression of dCas9 fused to transcriptional activators and repressors. This is an advance for the field because this CRISPRi/a system can now be used to study the function of genes essential for asexual growth. Furthermore, these tools could one day allow high-throughput screening efforts to investigate *P. falciparum* biology.

The effect of CRISPRi/a was then tested on several genes required for asexual growth, gametocytogenesis, and a gene with a putative role in drug resistance. The data are mostly well controlled and the authors should be commended for their comprehensive approach. The manuscript is well written and the conclusions are supported by the data.

Q1: Fig. 1C and D: How many times was this experiment performed? There are no statistics for the graph.

A1: We performed three biological replicates for this experiment, and we have summarized these three replicates and show the averages and standard deviations in Figure 1D.

Q2: Fig. 2B and D: No statistics are provided for the bar graphs. This is important to understand the relationship between gene knockdown and the observed phenotype.

A2: Thanks for your suggestion. We performed paired T-tests by comparing ΔCt (+Rap) to ΔCt (-Rap) (Please see details in the methods, lines 433-434 and 460-461) and showed the *P* values in the new Figures 3B and 3D after revision.

Q3: Fig. 3C and E: The statistics are missing again and it is not clear how many times the experiment was repeated. It would be useful to the reader to have this information in the figure legends.

A3: We performed three biological replicates for these experiments. As your suggestion, we added this information in the figure legends and the methods section. We also did the paired T-tests by comparing ΔCt (+Rap) to ΔCt (-Rap) and showed the *P* values in new Figures 4C and 4E after revision.

Q4: Fig. S2: The bar graphs have not been statistically analyzed. The number of biological repeats are not provided.

A4: We did the same statistical analysis described in **A2** and **A3**, and show the corresponding *P* values in new Figure 2. We also added the information of replicates (three) in the figure legends

of new Figure 2 after revision (Fig. S2 was moved to the main text to become new Figure 2 according to reviewer #2's suggestion).

Q5: Lines 176-181: There is no figure referenced for these results.

A4: Added accordingly (lines 183-186).

Reviewer #2

In the submitted manuscript, Liang et al. sought to develop a conditional CRISPR1/a system in the malaria parasite, *P. falciparum*, that allowed for a more controlled expression of dCas9. Thus, enabling the more accurate study of the effects of gene expression induction or knockdown to the parasite's growth and biology. The authors develop a DiCre recombinase inducible excision within *P. falciparum* to regulate the expression of either dCas9-GCN5 or -Sir2a. It is unclear whether the authors have achieved a fully controllable system and, while the level of gene expression changes, this system does not always produce the desired impact. While this study provides a new tool in the molecular arsenal for understanding malaria parasite biology, this system is still plagued by the fact that the results may still be influenced by the leaky expression of dCas9 and the resulting moderate levels of induction/repression of gene of interest. However, it will be useful when applied in conjunction with other tools already available.

Major Comments:

Q1: While the authors sought to create a non-leaky, leaky inducible system, they fail to address the low level of excision that occurs at 0h after induction (Figure 1D) and the lack of protein expression of dCas9-GCN5 (Figure 1E). Have the authors examined the stage specific effects of the excision and expression of dCas9 carefully throughout the IDC in replicate? Does induction of excision always take 48 hours to be effective or is the result presented just a stage-specific effect?

A1: The low level of excision that occurs at 0h after induction is the background. When using ImageJ to measure the band density in the agarose gel, the readout in the area with no DNA band always shows a background (not zero). To avoid any confusion, we made a new Figure 1D, in which we subtracted these backgrounds and shows no DNA content at 0h. Consistently, we added Western blot data in Figure 1B to show no and high expression of dCas9-GCN5/Sir2a before and after Rap induction, respectively. Figure 1E shows the protein expressions of dCas9-GCN5/Sir2a were gradually increased from early (Ring) through late time point (Schizont) after induction. No dCas9-GCN5 was detected in the Western blot at the early time point (R) probably because the expression level was too low to be detected by the Western blot.

Thanks for your suggestion on investigating whether induction of excision is stage-specific, we also examined excision events starting from trophozoite or schizont for 48h. The results show that there is no stage-specific excision during asexual development (see new Figure S2 and lines 149-150).

Q2: It is curious that the authors chose to place much of the data that evaluates the efficacy of their system in the Supplement. It would be preferable that the authors move the data in Figure S2 to the main text. While the results can be confusing and not as definitive one would want in a new tool, this is the reality of the system. The results presented here are more impactful than the current Figure 2.

A2: Agree with your comment. We moved Figure S2 to the main text, now it is new Figure 2.

Q3: The authors use the "extreme" example of a stage-specific gene, *ap2-g*, whose expression drives the conversion of the malaria blood stage parasite from asexual to sexual development. While it was interesting to see the various levels of impact each guide location had on the induction of expression via dCas9-GCN5, the level of induction doesn't compare to current systems that exist. Additionally, the use of *gexp5* as a "marker" of gametocyte gene upregulation is questionable because it is known to be expressed independent of AP2-G.

A3: The current conditional systems that exist in malaria parasites are normally for the knockdown of gene expression except that there was an excellent system for conditional activation of the *ap2-g* expression by the similar approach we used in this manuscript. That is the insertion of a *loxP* expression cassette (a strong constitutive calmodulin promoter, the *hdhfr* selectable marker, and a 3' UTR) just upstream of *ap2-g* (PMC7286680). Rap induced the DiCre excision of the *hdhfr* marker and 3' UTR, and let calmodulin promoter drive the expression of *ap2-g* at a high level and eventually led to the sexual conversion at an extremely high level (90%). Our activation of *ap2-g* by dCas9-GCN5 caused only ~1% conversion although this number was significantly higher than the one without RAP induction (DSMO). We added this comparison in the section of the discussion (lines 355-358).

We choose *gexp5* as one of the early gametocyte markers to verify that there was indeed a higher number of gametocytes generated after Rap induction. We did not want to elucidate whether *ap2-g* regulates the expression of *gexp5* although *ap2-g* Chip-seq showed a peak in the promoter of *gexp5* and knockdown of *ap2-g* resulted in downregulation of *gexp5* (PMC7083873). We revised the main text to specify the purpose of this experiment (line 277).

Minor Comments:

Q1: The authors state that there is no evidence of 'leaky' expression of dCas9-GCB5 or Sir2A in their system. This claim should be supported by evidence because the authors never directly compare the TetR-DOZI to the DiCre systems they have generated. The ratio of unexcised to excised in Fig 1D at time 0h suggests that the system may be a bit leaky. If so, is this detrimental to the parasite (with or without gRNAs, with or without RAP/aTc). Please include the Western Blot evidence in the supplemental and any additional evidence that the transgenic dCas9 parasite lines have the same/similar growth phenotype to the wild type.

A1: Please see **A1** to the major comments. In terms of growth phenotype, we did not find any noticeable growth changes of dCas9 parasite lines (before and after RAP induction) to the wild type. We stated this data in the main text (lines 150-152).

Q2: Could the authors comment further on why the TetR-DOZI system was abandoned? This induction may provide a functional tool for reversing any phenotypes from gene activation/repression.

A2: Yes, TetR-DOZI system is an excellent system for gene knockdown. However, we found that there was a clear leakage of dCas9 under the control of TetR-DOZI and we did not find a noticeable phenotype after we introduced *ap2-g* gRNAs into this system probably because the leaky expression caused the less magnification of induction compared to our DiCre/*loxP* system. We addressed this issue in the discussion (lines 305-308).

Q3: In Figure 1A, the authors illustrate the DiCre/*loxP* inducible system. For clarity, what does the blue box just following the second NLS (prior to PfGCN5/Sir2a) represent? Why did the authors choose to include dual nuclear localization signals?

A3: A triple glycine-serine protein domain linker (GS3) is inserted downstream of NLS to allow for an extended conformation and maximal flexibility of the epigenetic enzyme domain (lines 655-658). The reason for using dual nuclear localization signals was to efficiently guide the dCas9 to the parasite nuclei. Please see our previously published paper in PNAS for details (PMID: 30584102).

Q4: Could the authors comment as to why the excision in the GCN5 line is so much less efficient than in the Sir2a line?

A4: The data in Fig. 1D indeed suggest that the excision in the GCN5 line is less efficient than in the Sir2a line. However, these results were from one biological replicate of this experiment. We did two more replicates and the overall results from three replicates indicated the excision in the GCN5 line is similar to the Sir2a line (see new Fig. 1D).

Q5: The authors should clarify in their figure legends the number of biological/technical replicates and what the error bars represent. It is unclear how many times some of these experiments were replicated.

a. Please clarify if the excision ratio in Figure 1D was performed more than once. If not, provide a biological $n > 2$ for the excision ratio.

b. Please clarify the number of replicates in Figure S2C-F. Again, a biological $n > 2$ is recommended.

c. Lastly, please clarify the biological replicates of the Atg18 and AP2-G CRISPRi/a experiments.

A5: All the experiments shown in the above-mentioned figs were done with three biological replicates. We also performed statistical analysis according to reviewer #1's suggestion (Please see details in A2 and A3 to reviewer #1's comments).

Q6: To verify the expression of individual genes of interest, the authors extract RNA at 40-46hpi and compare +/-RAP parasites.

a. Why did the authors choose 40-46hpi when the most efficient excision occurs later than that according to Fig. 1D? Did the authors examine the levels of mRNA during the next cycle following RAP treatment?

b. I am curious about the effects of RAP treatment on the parasite progression. Have the authors compared developmental progression of the DiCre parasites in the presence of RAP with and without the addition of gRNAs. Have the authors determined the changes in expression measured by a single timepoint RT-PCR are not purely due to any developmental delays from off target dCas9 interactions?

A6: a. We chose this time-point (40-46hpi) because 1). the target genes express at a peak level at the late stage of the asexual development (new Figure 2A); 2). we wanted to investigate how conditional dCas9 regulates the target genes in the same cell cycle of RAP induction. We did measure the mRNA in the next cycle for some target genes with similar trends as the 40-46hpi. However, we were worried that targeting gene expression by our dCas9 system for a longer time (such as the next cycle after induction) will also impact the downstream gene expression including the internal control for RT-PCR. For example, dCas9 regulates *PfATG18* resulting in growth retardations after the first cycle of induction, making it difficult to estimate the mRNA levels in these sick parasites.

b. We did not see any noticeable changes in the developmental progression in the presence of RAP for a short time (2h) with and without the addition of gRNAs. Our early publication

showed the same results when dCas9-GCN5/Sir2a were constitutively expressed in the parasite with or without gRNA (PMID: 30584102).

Q7: Could the authors speculate why does asexual replication decrease in both Sir2a/GCN5 ATG18 line? Do the authors perform these growth assays in biological triplicate for each guide or just triplicate in total (once for each guide)? Is there a growth phenotype with +/RAP and no guides?

A7: ATG18 is an essential gene and can not be disrupted directly. The reason that both up-or down-regulation of this gene by dCas9-GCN5/Sir2a resulted in asexual replication decrease could be because ATG18 is critical for autophagy, food vacuole dynamics and apicoplast biogenesis, which may be involved in pivotal pathways (such as hemoglobin uptaking and digestion) in the malaria parasite. We discussed this in the section of discussion (lines 344-347). We did not see any noticeable growth phenotype in the presence of RAP without the addition of gRNAs.

Q8: While the authors provide evidence for expression changes (RT-PCR) in both the Sir2A/GCN5 lines, there are no statistics provided (Figure S2) and there is a lack of any growth phenotype (noted in text, but no evidence provided) for "essential" genes that were targeted. Can the authors comment whether this system will be useful considering these results?

A8: Thanks for your suggestion, we have provided the statistic information for all RT-PCR (Please see details in A2 and A3 to reviewer #1's comments). Although our conditional dCas9 system targeted several "essential" genes, only one target (ATG18) showed a growth phenotype whereas targeting other genes (such as K13, MYST, CHC, and GCN5) did not show a noticeable growth phenotype. However, we found that up-or down-regulation of K13 and MYST by dCas9-GCN5/Sir2a changed the sensitivities to Artemisinin (Please see details in new Fig. 2 and lines 34-35, 211-225, 352-355, 451-456).

Q9: The authors state that, "since this study aimed to build inducibility into the dCas9 system, we did not vigorously evaluate the phenotypic changes under other conditions." While I agree that the goal was to build an inducible system, isn't the utility of that system linked to the ability to result in a phenotype? As such, two recommendations that could potentially strengthen this study are to:

- a. Correlate an altered expression of atg18 or k13 to drug sensitivity
- b. Induce gene expression changes of a gametocyte-specific gene while evaluating the ability of RAP to induce recombinase activity in sexually developing parasites.

A9: a. Thanks for the suggestions, we tested the correlation between altered k13 or MYST expression and drug sensitivity (Please see details in new Fig. 2, sections of results and lines 34-35, 211-225, 352-355, 451-456).

b. This is an excellent suggestion. We performed more experiments and found that RAP could induce excision in gametocytes and ATG18 expression was up-or down-regulated in gametocytes by our system with ATG18 gRNA1. Upregulation of ATG18 by dCas9-GCN5 also led to a decrease in gametocyte growth (see details in new Fig. S3 and lines 40, 283-294, 397-401, 445-450).

Q10: There are a few clarifications that are necessary for the "Materials and Methods":

- a. Distinguish which strain of *P. falciparum* that is being used for the study and describe its ability to make mature gametocytes which are transmission-competent.
- b. Please confirm what stage the parasites are at 48h following RAP addition. This will establish whether the highest effects are in the same or next cycle.
- c. Please describe the numbers of replicates and controls for each experiment.

d. Describe the statistical analyses that were performed to generate p-values in some experiments.

A10: **a.** we used 3D7 strain in this study (line 384). 3D7 can produce healthy mature gametocytes for infection of mosquito (PMC3396512). **b.** in the next cycle. **c.** all the experiments were done with 3 biological replicates, averages and SDs were shown in all the figures. **d.** Done (see details in methods and figure legends).

April 18, 2022

Dr. Jun Miao
University of South Florida
Tampa

Re: Spectrum02782-21R1 (A leak-free inducible CRISPRi/a system for gene functional studies in *Plasmodium falciparum*)

Dear Dr. Jun Miao:

Thank you for addressing the reviewer comments so thoroughly.

Your manuscript has been accepted, and I am forwarding it to the ASM Journals Department for publication. You will be notified when your proofs are ready to be viewed.

Sincerely,

Björn Kafsack
Editor, Microbiology Spectrum
